# Structure of the *T. brucei* kinetoplastid RNA editing substrate-binding complex core component, RESC5

**Raul Salinas**☯, **Emily Cannistraci**[ID]☯, **Maria A. Schumacher**[ID]*

Department of Biochemistry, Duke University School of Medicine, DUMC, Durham, NC, United States of America

☯ These authors contributed equally to this work.
* maria.schumacher@duke.edu

**Data Availability Statement:** Coordinates and structure factor amplitudes for RESC5 have been deposited with the Protein Data Bank under the accession code 8DPK.

## Abstract

Kinetoplastid protists such as *Trypanosoma brucei* undergo an unusual process of mitochondrial uridine (U) insertion and deletion editing termed kinetoplastid RNA editing (kRNA editing). This extensive form of editing, which is mediated by guide RNAs (gRNAs), can involve the insertion of hundreds of Us and deletion of tens of Us to form a functional mitochondrial mRNA transcript. kRNA editing is catalyzed by the 20 S editosome/RECC. However, gRNA directed, processive editing requires the RNA editing substrate binding complex (RESC), which is comprised of 6 core proteins, RESC1-RESC6. To date there are no structures of RESC proteins or complexes and because RESC proteins show no homology to proteins of known structure, their molecular architecture remains unknown. RESC5 is a key core component in forming the foundation of the RESC complex. To gain insight into the RESC5 protein we performed biochemical and structural studies. We show that RESC5 is monomeric and we report the *T. brucei* RESC5 crystal structure to 1.95 Å. RESC5 harbors a dimethylarginine dimethylaminohydrolase-like (DDAH) fold. DDAH enzymes hydrolyze methylated arginine residues produced during protein degradation. However, RESC5 is missing two key catalytic DDAH residues and does bind DDAH substrate or product. Implications of the fold for RESC5 function are discussed. This structure provides the first structural view of an RESC protein.

## Introduction

*Trypanosoma brucei*, *T. cruzi* and *Leishmania* belong to the kinetoplastid clade within the phylum Euglenozoa and are the causative agents of African sleeping sickness, Chagas disease and leishmaniasis, respectively [1–3]. These flagellated protists are named for their unusual mitochondrial DNA, the kinetoplast, which is comprised of dozens of maxicircles and thousands of minicircles that are concatenated into a compacted network [1–3]. In addition to rRNAs, the mitochondrial maxicircles harbor 18 protein encoding genes, which primarily generate protein subunits of the respiratory chain [4]. However, for these mitochondrial mRNAs to be translated, a unique and extensive form of editing called kinetoplastid RNA editing (kRNA

**Funding:** This research was supported by Nanaline H Duke Endowed Chair and National Institutes of Health grants (R35GM130290 to M.A.S.). https://www.nigms.nih.gov. The ALS (Berkeley, CA) is a national user facility operated by Lawrence Berkeley National Laboratory on behalf of the US Department of Energy under Contract DE-AC02-05CH11231, Office of Basic Energy Sciences. Beamline 5.0.2 and 5.0.1 of the ALS, a US Department of Energy Office of Science User Facility under Contract DE-AC02-05CH11231, is supported in part by the ALS-ENABLE program funded by the NIH, National Institute of General Medical Sciences, Grant P30 GM124169-01. There was no additional external funding received for this study. The funders had no role in study design, data collection and analysis, decision to publish, or preparation of the manuscript.

**Competing interests:** The authors have declared that no competing interests exist.

editing) is required [5–11]. This form of editing, which is essential for protist viability, involves the insertion and deletion of uridines (U) within the transcript to generate translatable mitochondrial mRNAs [5–11]. Most of the protein coding transcripts are heavily edited, termed pan editing [5–8]. Pan editing involves multiple insertion and deletion steps and is mediated by several gRNAs within one mRNA transcript. This form of editing is so extreme that hundreds of Us may be inserted and tens of Us deleted, leading to transcripts that are often double the size of the pre-edited mRNA. The editing process is catalyzed by large, multiprotein editosome/RNA editing core complexes (RECC) and directed by guide RNA (gRNAs), the latter of which are primarily encoded on minicircle DNA [12, 13].

Interestingly, the editosome/RECC, although responsible for performing the catalytic steps, only permits a single step of editing [14]. Indeed, the process is highly dynamic and studies revealed that complete kRNA editing requires accessory factors, which mediate RNA interactions and processivity [6, 15–36]. Most of these accessory factors show no homology to structurally characterized proteins and because of the essentiality of kRNA editing, these factors have been suggested as attractive targets for specific chemotherapeutic design. Early studies identified several accessory factors involved in editing that were later shown to be required for editing single mRNA transcripts such as MRP1/MRP2, p22 and MRB1590 [16, 18, 21]. Subsequent work uncovered a macromolecular complex consisting of six core proteins that plays an essential role in processive editing of transcripts and which is also responsible for gRNA binding and stability [20, 32, 33, 36]. This complex, which was simultaneously discovered by the Stuart, Aphasizhev and Lukeš lab, was called the RNA-editing substrate-binding complex (RESC) and is now known to serve as the central platform for RNA editing [20, 32, 33, 36]. Further underscoring the importance of this complex, recent data indicates that the RESC also acts as a hub for the coordination of editing with other RNA processing events such as mRNA maturation and translation [7, 8, 11]. Two hybrid and pulldown studies in the Read laboratory revealed information about the assembly of the RESC complex including that it is dynamic and composed of six core subunits, called RESC1-RESC6, that interact in an RNA-independent manner [26]. Interestingly, none of the RESC core proteins show homology to proteins of known structure and there is currently no structural information for any RESC component. More recent work has indicated that the RESC may be larger with as many as 13 subunits, in addition to the core six components [37].

Of the six core RESC proteins, RESC1 and RESC2 (formerly called GAP1/GRBC1 and GAP2/GRBC2) form a heteromeric complex, RESC1-RESC2, that is responsible for gRNA binding and stabilization; *gap* knockouts lead to destabilization and loss of all gRNAs and hence such knockouts abrogate editing [20, 32, 33, 36]. Similar knockdown studies revealed essential roles for RESC5 (formerly called MRB11870) in kRNA editing; cells depleted of RESC5 exhibited 80–90% decrease in edited mRNAs and 2–4 fold increases in pre-edited mRNAs [25, 27, 28]. In addition, the knockdown cells also showed defects at 3′ most editing sites suggesting a failure in the initiation of editing in these cells as editing proceeds from a 3′ to 5′ direction within a transcript [25, 27, 28]. Pulldown experiments indicate that RESC5 appears to be a key part of the foundation for the RESC complex, as downregulation of the proteins dramatically affected interactions within the RESC core [25, 27, 28].

RESC5 contains 310 residues and like other RESC proteins, shows no homology to structurally characterized proteins. Thus, to gain insight into this RESC component we performed structural and biochemical studies. We describe here the 1.95 Å structure of the *T. brucei* RESC5 protein, which represents the first report of an RESC protein structure. Our data show that RESC5 is a monomeric protein. Surface analyses of the RESC5 structure highlight potential binding sites for proteins and RNA. Structural homology searches revealed that the RESC5 structure harbors a fold that is similar to that found in the dimethylarginine

dimethylaminohydrolase (DDAH) enzymes. However, RESC5 is missing key DDAH catalytic residues and hence the RESC5 fold appears to have adapted to a distinct function [38–43]. Thus, these data reveal the high-resolution structure of a central component required for kRNA editing in kinetoplastids.

## Materials and methods

### Purification of *T. brucei* RESC5 proteins

Multiple sequence alignments revealed that RESC5 residues 1–6 are not conserved and are predicted to be disordered (using the GOR4 server) [44]. Hence, a synthetic gene, encoding RESC5 residues 7–310, that was codon optimized for expression in *E. coli*, was obtained from Genscript and subcloned into the NdeI and XhoI sites of the pET15b expression vector. RESC5(7–310) was, however, insoluble under all conditions tested and hence pure protein could not be obtained. Because RESC5 C-terminal residues 287–310 were also predicted to be largely composed of coil regions or disordered (using the GOR4 server) [44] and are also not conserved among Trypanosome RESC5 homologs, RESC5(7–286) was generated and was found to be expressed at high levels in soluble form. An RESC5(R71A-P180H-A277C) synthetic gene (encompassing residues 7–286) was also generated, codon optimized for *E. coli* expression and subcloned into the NdeI and XhoI sites of the pET15b expression vector. Expression of both RESC5(7–286) proteins results in the addition of an N-terminal hexa-histidine tag for purification. Constructs were transformed into *E. coli* C41(DE3) cells. For protein expression, RESC5(7–286) and the RESC5 mutant plasmid containing cells were grown to an $OD_{600}$ of 0.5–0.6 and induced with 1 mM IPTG at 15°C overnight. For purification of both the WT and mutant RESC5(7–286), cells were lysed in buffer A (25 mM Tris pH 7.5, 300 mM NaCl, 5% (v/v) glycerol, 1 mM β-mercaptoethanol (βME)) using a microfluidizer and cell debris removed by centrifugation at 34,900 xg. The lysate was loaded onto a Cobalt NTA column and the column washed with increasing concentrations of imidazole in buffer A. Proteins were eluted in batch mode with 30, 40, 50, 75, 100, 200, 300 500 mM imidazole fractions and samples containing the protein were concentrated prior to loading onto an S75 size exclusion chromatography (SEC) column for final purification. Pure fractions were combined and concentrated for biochemical and crystallographic studies.

### Crystallization and structure determination of RESC5(7–287)

Purified RESC5(7–287) was concentrated to 15 mg/mL and utilized in hanging drop vapor diffusion screens at room temperature (rt). For these screens, Wizard screens I-IV, PegRx1, PegRx2 and Cryo screens I and II were utilized. Crystals were produced in conditions containing high molecular weight PEGs (PEG 4000, PEG 6000 and PEG 8000) and LiSO$_4$. Optimal crystals were obtained by mixing the protein 1:1 with a solution consisting of 20% PEG 6000, 0.1 M Imidazole pH 8.0, 0.2 M LiSO$_4$ and took one week to reach maximum size. The crystals were cryo-preserved by looping and dipping a crystal for 1–2 s in a solution containing the crystallization reagent supplemented with 25% (v/v) glycerol. Data were collected at the Advanced Light Source (ALS) beamline 5.0.2 and processed with XDS [45]. Native data were collected to 1.95 Å and to obtain phases, crystals were soaked for 3 days in HgCl$_2$. A Mercury SAD data set was collected to 2.3 Å resolution and used to obtain an initial set of phases in Phenix_Autosol [46]. There are two subunits in the crystallographic asymmetric unit (ASU). After a partial structure was constructed using Coot [47], Phenix refinement [46] commenced resulting in significantly improved map allowing for residues 7–285 of each subunit to be constructed. For final refinement the high-resolution native data set was utilized. Using these data, the structure was refined to final R$_{work}$/R$_{free}$ values of 22.1%/25.7% to 1.95 Å resolution.

## Thermal shift assays

Thermal shift assays were performed using Bio-Rad CFX Connect Detection System. For these assays 10 μM of RESC5 WT and mutant proteins were analyzed in the absence and presence of varying concentrations of L-citruline (AdooQ Bioscience) or dimethylamino arginine (AdooQ Bioscience). The RESC5 proteins and all other reagents used in this experiment were diluted in filter sterilized low salt buffer (25 mM Tris pH 7.5, 150 mM NaCl, 5% (v/v) glycerol, and 1mM β-ME). Melting curves were collected for 10 μM of RESC5 (WT or mutant), in the presence of 0 μM, 100 μM, 200 μM, 500 μM and 1 mM of L-citrulline or dimethylamino arginine. The appropriate volumes of RESC5 protein, L-citrulline, or dimethylamino arginine for each condition described above were added to wells of a 96 well Bio-Rad Hard-Shell Plate (thin walls) containing a final volume of 20 μL and 1X of GloMelt (GloMelt$^{TM}$ Thermal Shift Protein Stability Kit from Biotium-Cat No. 33021–1). The plates were briefly spun before to transfer to the Bio-Rad CFX Connect Detection System. Fluorescence was detected over a temperature range of 25–100˚C with 0.5˚C steps and a time hold of 1 min for each temperature step.

## Size exclusion chromatography (SEC) analyses

SEC studies were carried out on RESC5 at 2.7 mg/mL. The protein was injected onto a Superdex S75 column (Fisher) with a mobile phase of 50 mM Tris pH 7.5, 150 mM NaCl, 5% (v/v) glycerol. The RESC5 elution volume was compared to a series of protein standards to determine the molecular weight. The standards were aprotinin (6.5 kDa), cytochrome c oxidase (12 kDa), carbonic anhydrase (29 kDa), and albumin (66 kDa).

## Bioinformatic analyses of phylogenetics

For bioinformatic analyses of DDAH proteins, DDAH sequences were searched for and analyzed using the UniPort database. RESC5 homologs were searched for using NCBI protein Blast. RESC5 homologs were only found in kinetoplastids. Hence a FASTA file containing 22 RESC5 were downloaded from a variety of kinetoplasidae. No DDAH homologs were identified in the eukaryotic phyla Annelida, Chidaria, Echinodermata, Mollusca, Porifera, Ctenophora, Rotifera and Nematodes. FASTA files from chordates, eubacteria, and arthropods were combined, then this combined FASTA file and the FASTA file for kinetoplastidae were fed into Clustal Omega to perform sequence alignments. Alignment files were obtained from Clustal Omega then aligned sequences were analyzed in Jalview 2.112.4. Jalview was used to calculate evolutionary distances between the aligned sequences using Neighbor Joining and BLOSUM62. This tree was saved as a Newick file then fed into PhyloT to generate a phylogenetic tree. iTol was used to visualize and annotate the phylogenetic tree [48].

## Results

### Overall crystal structure of *T. brucei* RESC5

Studies have shown that downregulation of RESC5 dramatically impacts stabilization and formation of the RESC core [27]. To gain insight into the structure and function of RESC5 we performed biochemical and structural studies. We first characterized the oligomeric state of RESC5 as to date there is little information as to the oligomerization status of the components within the RESC core. Size exclusion chromatography (SEC) studies revealed a clear, single peak for the purified RESC5 protein with an estimated molecular weight (MW) of 30 kDa, consistent with a monomeric form of the protein (which has a predicted MW of 32 kDa) (Fig 1A and 1B; S1 Fig). We next performed crystallization trials on the RESC5 protein. Crystals that diffracted to high resolution were obtained and the structure solved by mercury single

**A**

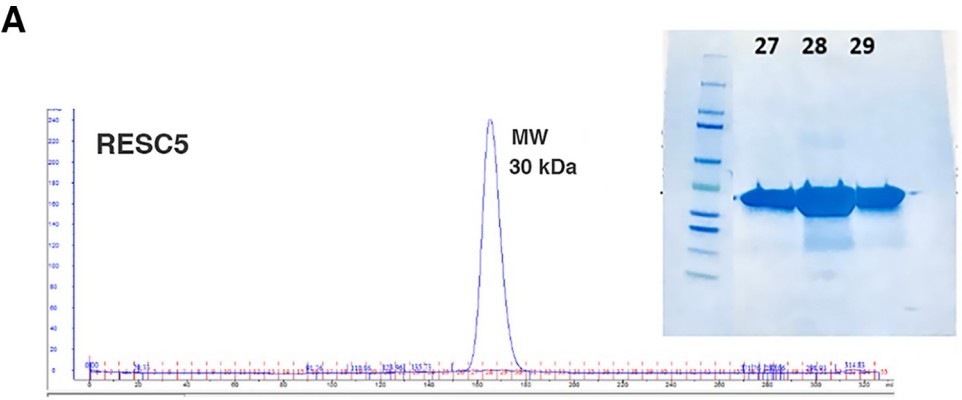

**B**

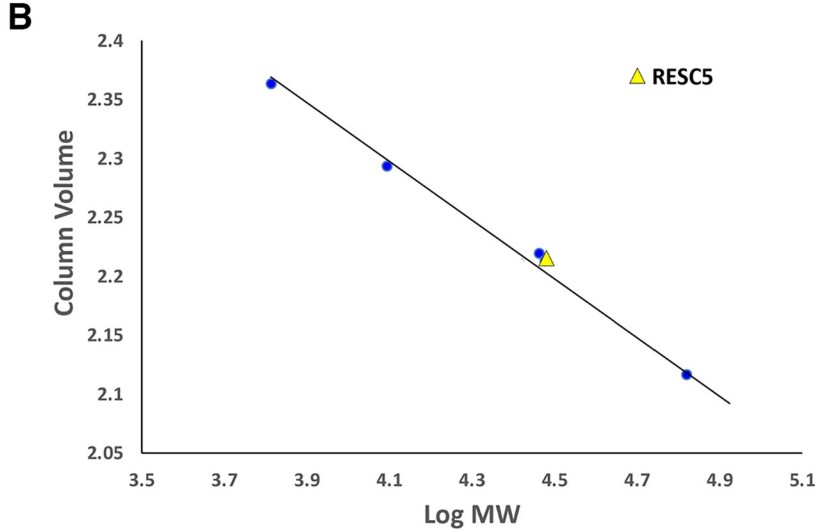

**Fig 1. Size exclusion chromatography (SEC) analyses of RESC5 reveals monomer. A** SEC analysis of the *T. brucei* RESC5. The SEC profile is shown. **B** SEC analyses of MW. The x and y axes are Log MW and column volume, respectively. RESC5 eluted at calculated MWs of 30 kDa (yellow triangle). The standards used for calculation of the standard curve are shown as blue diamonds and were aproptinin (6.5 kDa), cytochrome c oxidase (12.4 kDa), carbonic anhydrase (29 kDa), and albumin (66 kDa).

wavelength anomalous diffraction (SAD) to 1.95 Å resolution (Table 1) (Fig 2; S2 Fig). There are two RESC5 subunits in the crystallographic asymmetric unit (ASU). PISA analysis shows that the largest buried surface area (BSA) between subunits is only ~240 Å$^2$ [49], consistent with our SEC data indicating that the protein is monomeric. The two monomers in the ASU superimpose with a root mean square deviation of 0.6 Å for 274 Cα atoms indicating they adopt the same structure. The only differences are in two surface exposed loops (residues 18–34 and 159–167), which appear flexible. Hence, we will limit the discussion to a single monomer. The RESC5 structure is composed of one domain and harbors a propeller-like fold comprised of 13 β-strands and 10 helices (Fig 2). The topology is (β1: residues 11–17; α1: 35–51; β2: 57–61; α2: 67–75; β3: 76–79; β4: 82–85; α3:95–108; α4: 116–120; α5: 127–130; β5: 131–134; β6:138–142; α6: 148–158; β7:168–173; α7:181–184; β8:185–188; β9:192–196; α8:199–211; β10:218–222; β11:239–243; α9:247–256; β12:259–263; α10: 266–271; β13: 280–285). In the

**Table 1. Data collection and refinement statistics: *T. brucei* RESC5.**

| Protein | RESC5 |
|---|---|
| **Data collection** | |
| Pdb code | 8DPK |
| Space group | P2$_1$ |
| Cell dimensions | |
| *a, b, c* (Å) | 45.6, 80.5, 76.1 |
| α, β, γ (˚) | 90.0, 102.7, 90.0 |
| Resolution (Å) | 54.52–1.95 |
| | (2.01–1.95)$^a$ |
| R$_{sym}$ | 0.070 (0.274) |
| R$_{pim}$ | 0.044 (0.228) |
| I/σI | 7.4 (1.0) |
| Completeness (%) | 99.1 (89.0) |
| Redundancy | 3.0 (2.1) |
| CC(1/2) | 0.994 (0.673) |
| **Refinement** | |
| Resolution (Å) | 54.52–1.95 |
| No. reflections | 38130 (2269) |
| R$_{work}$/R$_{free}$ (%) | 22.1/25.7 |
| R.m.s. deviations | |
| Bond lengths (Å) | 0.004 |
| Bond angles (˚) | 0.623 |
| Ramachandran analyses | |
| Favored (%) | 97.4 |
| Disallowed (%) | 0.0 |

$^a$Values in parentheses are for highest-resolution shell.

structure there are five similar modular "blades" with each containing a two or three-stranded β-sheet packed against an α-helix (Fig 2).

## RESC5 harbors a propeller-fold with homology to dimethylarginine dimethylaminohydrolases

DALI searches revealed the RESC5 structure shows the most significant structural similarity to proteins belonging to the superfamily of arginine-glycine amidotransferases (3.75.10.10) [38–43], with the strongest homology to dimethylarginine dimethylaminohydrolases (DDAH) of this superfamily. Dimethylarginine residues, which are produced during protein degradation, are potent inhibitors of nitric oxide synthase (NOS). Hence, hydrolysis of dimethylarginines by DDAH enzymes enhances the production of nitric oxide (NO) while inhibition of DDAH activity blocks NO synthesis leading to vasoconstriction [38–43]. Superimposition of 247 Cα atoms of RESC5 onto the corresponding Cα atoms of bovine DDAH (pdb code: 2CI6), using the secondary-structure matching (SSM) program [50] results in an rmsd of 2.4 Å and a Z-score of 13.0 (Fig 3A). The RESC5 structure also shows weak homology to the structure of the ribosome anti-association factor IF6 (pdb code: 1G61), another β-propellor fold family member; the two structures can be overlaid in SSM, resulting in an rmsd of 3.5 Å and a Z-score of 5.8 for 165 Cα atoms (S3 Fig) [50, 51]. Interestingly, IF6 proteins function in protein-protein interactions as does RESC5, with IF6 forming a complex with the large ribosomal subunit. In

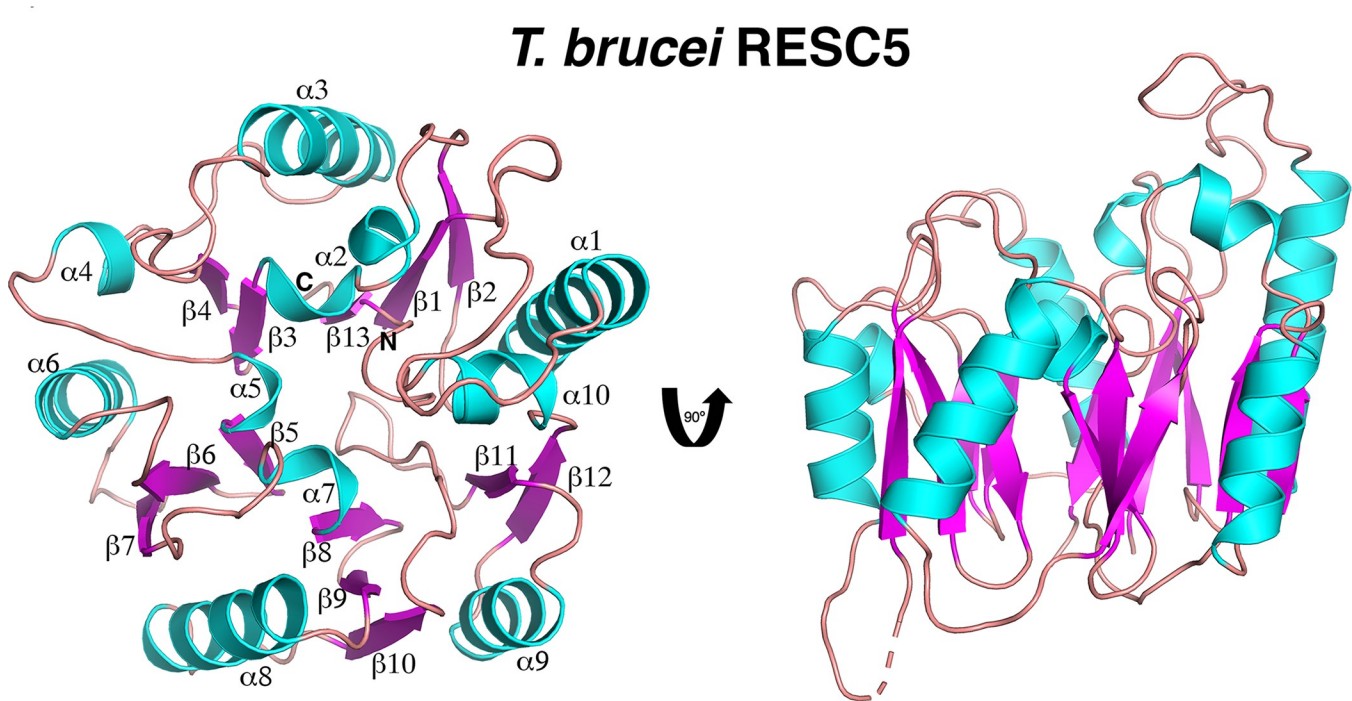

**Fig 2. Crystal structure of *T. brucei* RESC5.** Cartoon figure of RESC5 with strands colored magenta and helices, cyan. Shown are two views of the structure related by a 90˚ rotation about the y axis. Secondary structure elements are labeled in the view on the left. All ribbon diagrams were made using PyMOL [58].

this complex, IF6 inhibits the interaction of the large subunit with the small subunit, thus regulating translation [52, 53].

As noted, however, the strongest structural homology detected with RESC5 in the protein data bank are to DDAH enzymes, which hydrolyze dimethylarginine substrates. Indeed, an unbiased search (CavityPlus) for the presence of pockets in the RESC5 structure [54] revealed the location corresponding to the active site in DDAH enzymes as a cavity in RESC5 (S4 Fig). The catalytic function of DDAH enzymes have been well studied and revealed that the active site is composed of a conserved Cys-His-Glu triad, with the cysteine residue functioning as the nucleophile [39, 41]. Mutagenesis studies showed that the active site cysteine is essential for DDAH activity as substitution of this residue to alanine essentially abrogated DDAH catalysis [55].

In addition to the Cys-His-Glu triad, the DDAH active site contains several conserved residues important for substrate binding [55]. Examination of the superimposition of RESC5 onto DDAH revealed that the catalytic cysteine and histidine are not conserved in RESC5 and are replaced by alanine and proline residues, respectively (Fig 3B). In addition, the pockets show differences in overall architecture with some helices in DDAH replaced by loop regions in RESC5 (Fig 3A and 3B). However, several pocket residues are conserved between RESC5 and DDAH. For example, Asp73, Asp78, Arg98 and Arg145 in DDAH (pdb code: 2CI6) correspond to Glu68, Asp74, Arg94 and Arg145 in RESC5 (Fig 3B and 3C).

The conservation of some pocket residues between RESC5 and DDAH prompted us to ask if RESC5 could bind DDAH substrates or the DDAH product L-citrulline. To address this question, we employed thermal shift assays (Materials and methods) and analyzed the ability of RESC5 to interact with L-dimethylarginine or citrulline. The control experiment with just RESC5 revealed a thermal shift of 39˚C. Data collected on RESC5 in the presence of citrulline or dimethylarginine using concentrations up to 1 mM revealed no interaction of either

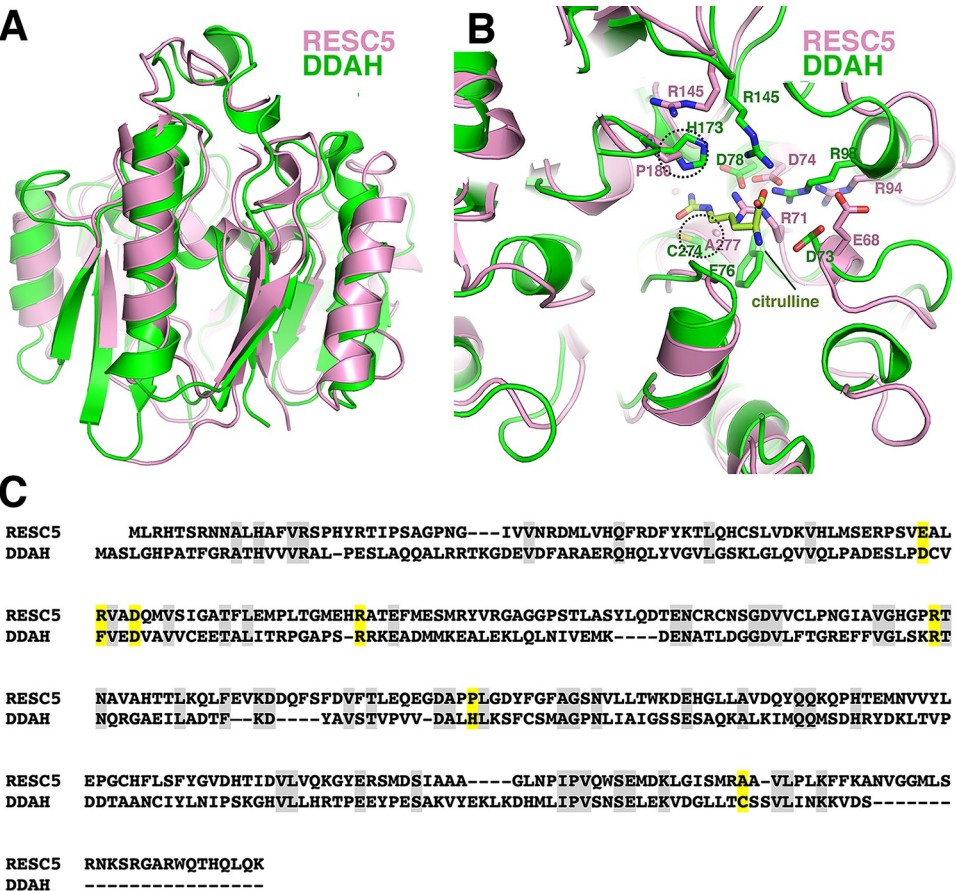

**Fig 3. _T. brucei_ RESC5 has a DDAH fold. A** Overlay of _T. brucei_ RESC5 (pink) and bovine DDAH (pdb code: 2CI6) structures (green). The overlay shows that although the positioning of secondary structural show some differences, they both contain the same overall fold. **B** Close up of the overlay from Fig 3A showing positions of key active site residues in DDAH. Also shown as sticks and labeled is the citrulline bound in the DDAH structure. While important arginine and aspartic acid residues in the active sites are conserved between the two proteins, central catalytic residues from the catalytic triad Cysteine-Histidine are Alanine-Proline in RESC5. **C** Structure based sequence alignment of _T. brucei_ RESC5 with DDAH. DDAH used is pdb id code: 2CI6 (from the alignment shown in Fig 3A). In the sequence alignment conserved residues between the two proteins are colored grey. Yellow residues are those involved in catalysis or substrate/product binding by DDAH. Note, several of these residues are conserved in RESC5. However, notably the catalytic cysteine and histidine (asterisks) are not.

citrulline or L-dimethylarginine for RESC5; no change in the thermal shifts were observed upon the addition of these compounds (Fig 4A). To assess if replacement of RESC5 Ala277 and Pro180 with the corresponding residues in DDAH enzymes would impact interaction with citrulline or L-dimethylarginine we generated a RESC5 mutant. In addition to the P180H-A277C substitutions, we substituted the RESC5 Arg71 with alanine as examination of the binding pocket showed the presence of the arginine at position 71 would be predicted to clash with the substrate/product (Fig 3B). The R277A substitution was chosen instead of the phenylalanine found in DDAH enzymes because RESC5 residue Ile273 directly abuts the 277 position and would clash with a phenylalanine; this does not occur in DDAH enzymes because the corresponding region adopts a different structure in the DDAH enzyme (S5 Fig). The RESC5(R71A-P180H-A277C) was purified and utilized in thermal shift assays as per the WT RESC5. These experiments revealed that the mutant RESC5 protein had a higher melting temperature, however there was no evidence of the mutant interacting with citrulline or L-

**A**

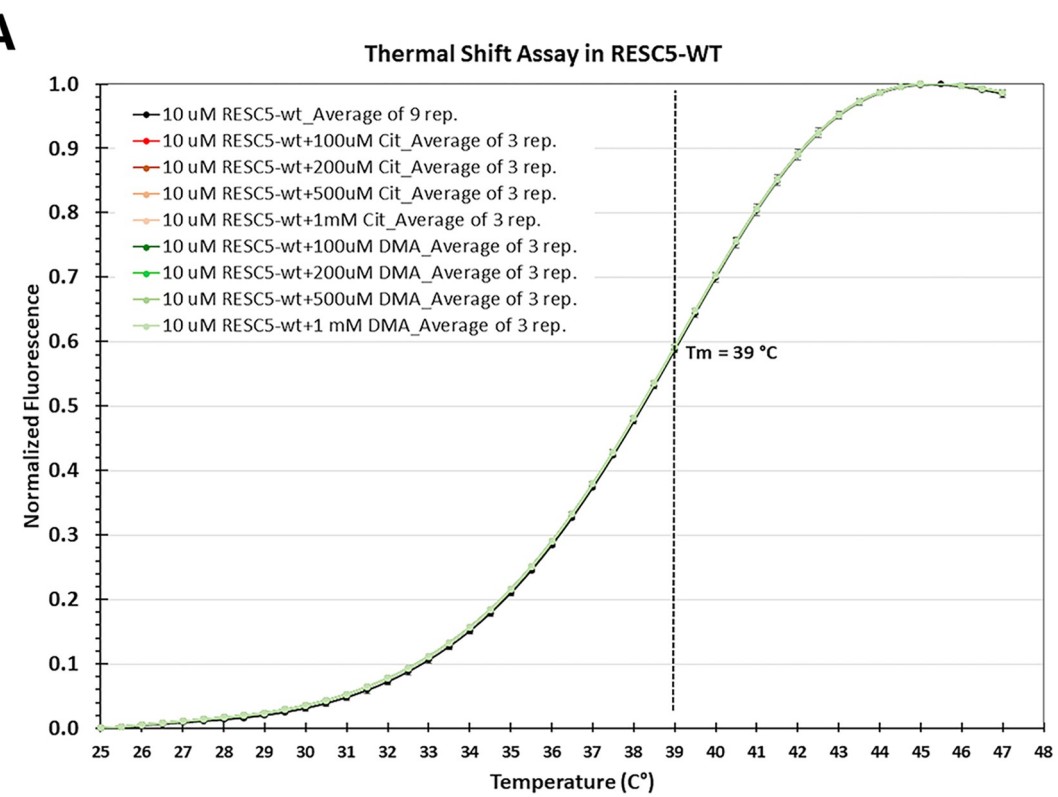

**B**

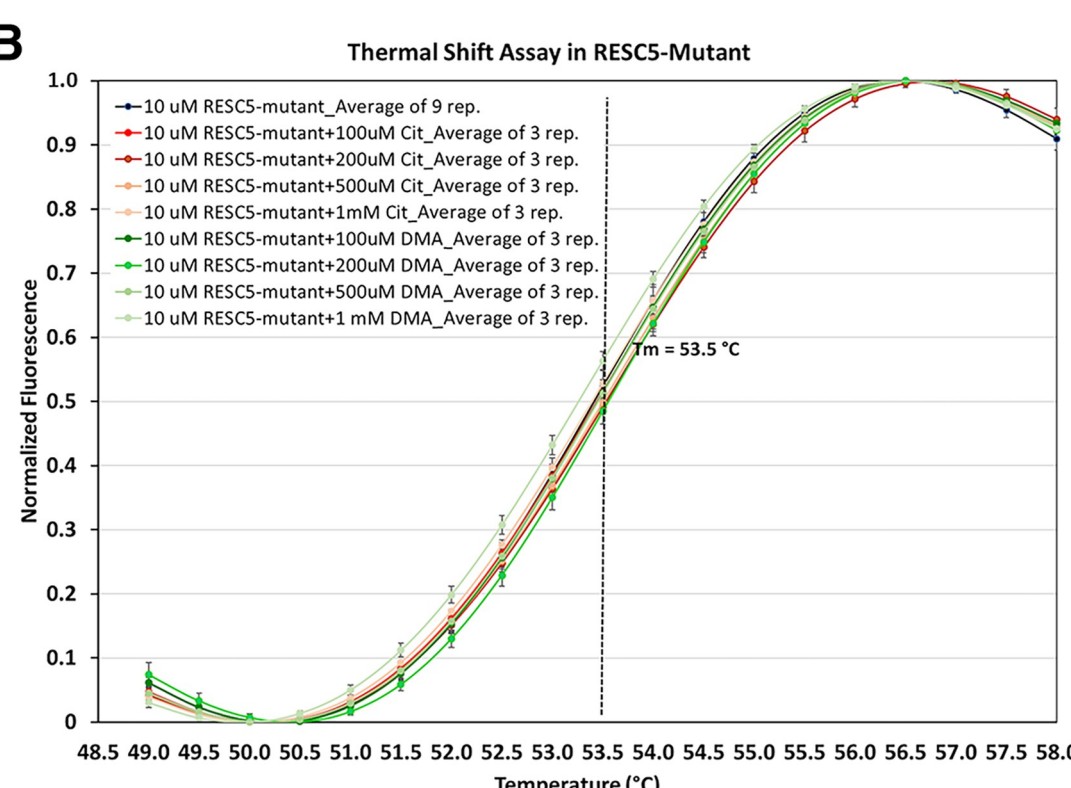

**Fig 4. Thermal shift assays analyzing RESC5 binding to DDAH substrate and product. A** Melting curves for 10 μM WT RESC5 in the presence of varying concentrations of L-citrulline and dimethylaminoarginine. Data was scaled from 0–1. **B** Melting curves for 10 μM RESC5(R71A-P180H-S277C) in the presence of varying concentrations of L-citrulline and dimethylaminoarginine. Data was scaled from 0–1. Error bars represent standard deviations from the average (of at least three technical triplicate runs).

dimethylarginine (Fig 4B). This may be explained by the noted difference in overall pocket architecture between RESC5 and DDAH.

## Surface properties of RESC5: Implications for protein-protein and protein-RNA interactions

RESC5 forms a key core component of the RESC complex as data indicates that it interacts with RESC6 and the RESC1-RESC2 subcomplex [26]. The RESC complex appears conserved among trypanosomes. Hence, we performed a sequence alignment of RESC5 homologs from Trypanosome species and utilized the alignment to map conserved residues on the RESC5 surface to gain insight into potential regions that may be involved in protein-protein interactions (Fig 5A). These analyses revealed several large patches of conserved residues on the RESC5 surface that may be involved in RESC complex formation (Fig 5B). In addition, while studies have demonstrated that RESC1 and RESC2 are the components of the RESC that mediate interactions with gRNAs, other proteins in the complex may participate in RNA interactions. Indeed, RNase treatment of RESC and accessory components leads to reduced interactions between some factors [26]. Thus, we analyzed the RESC5 electrostatic surface potential (Fig 6A and 6B). There are two major extended and exposed basic patches on RESC5, which may be available for nucleic acid interaction (Fig 6A and 6B). Interestingly, the largest patch corresponds to the location of the active site in DDAH enzymes suggesting this pocket might be involved in RNA binding. This region is also notably conserved among RESC5 proteins and hence could participate in both RNA and/or protein interactions (Fig 5B).

## Phylogenetic analyses of RESC5 and DDAH enzymes

The structural similarity of *T. brucei* RESC5 to DDAH enzymes led us to ask if DDAH proteins are found in Trypanosomes and other kinetoplastids and also whether RESC5-like proteins are present in other organisms. Notably, when we searched for RESC5 homologs in other organisms, the only clear RESC5 proteins we identified were from the class kinetoplastida, including the genus *Trypanosoma* and *Leishmania* in which the RESC complex has been biologically characterized (Fig 7A). Phylogenetic analyses of the DDAH enzyme revealed that homologs are widespread in higher eukaryotes but absent in the eukaryotic phyla Annelida, Chidaria, Echinodermata, Mollusca, Porifera, Ctenophora, Rotifera and Nematodes (Fig 7B). Most notably, no DDAH homologs were identified in kinetoplastids. Hence, whether the two protein folds (RESC5 and DDAH) are the result of a convergent evolution process or whether they may be derived from a protein with a similar fold is unclear. It will be of interest to further decipher the functions of RESC5 to see if it harbors capabilities to bind substrates within the DDAH-like pocket or whether it performs roles completely unrelated to the DDAH proteins.

## Discussion

Uridine insertion/deletion RNA editing is an essential and unique process that takes place in kinetoplastids and is required to create translatable open reading frames in most mitochondrially-encoded mRNAs. While the catalytic editosome/RECC is the catalytic machine that mediates editing steps, accessory proteins are also required for editing. The RESC complex has

**A**

β1　α1　β2　α2　β3

```
XP_823393.1      1   MLRHTSRNNALHAFVRSPHYRTIPSAGPNGIVVNRDMLVHQFRDFYKTLQHCSLVDKVHLMSERPSVEALRVADQMVSIG   80
RHW69169.1       1   MLRHTSRNNALHAFVRSPHYRTIPSAGPNGIVVNRDMLVHQFRDF-KTLQHCSLVDKVHLMSERPSVEALRVADQMVSIG   79
KAG8344931.1     1   MLHSTLVSKALHAFVRSPHYRSIPSAGPNGVVVNRDMLIHQFRDFYKTLQHCSLVDKVHLMSERPSVEALRVADQMVSIG   80
KAH9578299.1     1   MFRRTSGNKALHAFVRSPHYRSVPSAGPNGIVINRDMLIHQFRDFYKTLQHSSLVDKVHLMSERPSVEALRVADQMASIG   80
XP_028884646.1   1   MLRRTSGNKALHAFVRSPHYRSVPSAGPNGIVINRDMLIHQFRDFYKTLQHSSLVDKVHLMSERPSVEALRVADQMASIG   80
XP_029237384.1   1   MLRRTSGNKALHAFVRSPHYRSVPSAGPNGIVINRDMLIHQFRDFYKTLQHSSLVDKVHLMSERPSVEALRVADQMTSIG   80
XP_811845.1      1   MLRRTSGNKALHAFVRSPHYRSVPSAGPNGIVINRDMLIHQFRDFYKTLQHSSLVDKVHLMSERPSVEALRVADQMASIG   80
KAG5495086.1     1   -MKVTRSARAVHAFVRMPHHRSVPPTGPSGIIVNRDVLFRQFRDFYKTLQHSTLVDKVHLMTERPGVESLRVADQMTSLG   79
XP_010702224.1   1   -MRVTRSARAVHAFVRMPHHRSVPPTSPSGIIVNRDVLRQFRDFYKTLQHCTLVDKVHLMAERPGVESLRVADQMASLG   79
XP_015655064.1   1   -MKSTASLRALHAFVRAPHYRSIPPSGPSGIIINRDVLNRQFRDFYKTLQHSTLVDKLHLMAERPGVESMRVADQLACVG   79
```

β4　α3　α4　α5　β5　β6　α6

```
XP_823393.1      81  ATFLEMPLTGMEHRATEFMESMRYVRGAGGPSTLASYLQDTENCRCNSGDVVCLPNGIAVGHGPRTNAVAHTTLKQLF   158
RHW69169.1       80  ATFLEMPLTGMEHRATEFMESMRYVRGAGGPSTLASYLQDTENCRCNSGDVVCLPNGIAVGHGPRTNAVAHTTLKQLF   157
KAG8344931.1     81  ATFLGMPLTGMEHRATEFMESMRYVRGSGGPTTLASYLQDIENCRCHSGDIVCLPNGIAVGHGPRTNAVAHATLKQLF   158
KAH9578299.1     81  ATFLGMPLTGMEHRATEFMESMRYVRGAGGPSTLASYLQDTENCRCHSGDIVCLPTGIAVGHGPRTNAVAHATLKELF   158
XP_028884646.1   81  ATFLGMPLTGMEHRATEFMESMRYVRGAGGPSTLASYLQDTENCRCHSGDIVCLPTGIAVGHGPRTNAVAHATLKELF   158
XP_029237384.1   81  ATFLGMPLTGMEHRATEFMESMRYVRGAGGPSTLASYLQDTESCRCHSGDIVCLPTGIAVGHGPRTNAVAHSTLKQLF   158
XP_811845.1      81  ATFLGMPLTGMEHRATEFMESMRYVRGAGGPSTLASYLQDTESCRCHSGDIVCLPTGIAVGHGPRTNAVAHSTLKQLF   158
KAG5495086.1     80  PVLLGMPLTGMEHRSSEFLETMRYVRGAGGPTTLSAYLQDNDACRCHSGDIVCLPQGVAVGHGPRTNTVTHQLLRALF[2]  159
XP_010702224.1   80  PVLLGMPLTGMEHRSSEFLETMRYVRGAGGPTTVSAYLQDNDACRCHSGDIVCLPQGVAVGHGPRTNAAAHQVLRELF[3]  160
XP_015655064.1   80  PVLLGMPLTGMEHRATEFHDAMRYVRVAGGPTNVSPYLQDNDACRCHSGDIVVLPQGIAVGHGPRTNAATHQVLRDIF[9]  166
```

β7　α7　β8　β9　α8　β10

```
XP_823393.1      159  EVKDDQF   SFDVFTLEQEGDAPPLGDYFGFAGSNVLLTWKDEHGLLAVDQYQQKQPHTE--MNVVYLEPGCHFLSFY   232
RHW69169.1       158  EVKDDQF   SFDVFTLEQEGDAPPLGDYFGFAGSNVLLTWKDEHGLLAVDQYQQKQPHTE--MNVVYLEPGCHFLSFY   231
KAG8344931.1     159  EVKDDQF   AFDVFTLEQEGDAPPLGDYFSFAGNNVLLTWKDEHGLLAVDQYQQKQPNTE--MNVVYLEPGCHFLSFY   232
KAH9578299.1     159  EVKDDQF   SFDVFTLEQEGDAPPLGDYFGFAGNNVLLTWKDEHGLLAVDQYQQKQPHSE--MNIVYLEPGCHFLSFY   232
XP_028884646.1   159  EVKDDQF   SFDVFTLEQEGDAPPLGDYFGFAGNNVLLTWKDEHGLLAVDQYQQKQPHSE--MNIVYLEPGCHFLSFY   232
XP_029237384.1   159  EVKEDQF   TFDVFTLEQEGDAPPLGDYFGFAGNNVLLTWKDEHGLLAVDQYQQKQPNAE--MNVVYLEPGCHFLSFY   232
XP_811845.1      159  EVKEDQF   SFDVFTLEQEGDAPPLGDYFGFAGNNVLLTWKDEHGLLAVDQYQQKQPQVE--MNVVYLEPGCHFLSFY   232
KAG5495086.1     160  DAADGSI[15]KFEVVTLEQEGDAPPLGDYFGFAGNDILLVWKDEHGLLAVDQLQQQLAKSEkqFKVLYLEPGCHFLTFY   250
XP_010702224.1   161  DAPDGGT[15]KFEVVTLEQEGDAPPLGDYFGFAGNDILLVWKDEHGLLAVDQLQQQLAKSElqLQVLYLEPGCHFLTFY   251
XP_015655064.1   167  EAAEEGV[ 5]SFEVVTLEQEGDAPPLGDYFGFAGNNIILLVWKDEHGLLAVDQLQQQLAKTNqqLKVVYLEPGCHFFTFY   247
```

β11　α9　β12　α10　β13

```
XP_823393.1      233  GVDHTIDVLVQKGYERSMDSIAAAGLNPIPVQWSEMDKLGISMRAAVLPLKFFK---ANVGGMLSRNKSRGARWQTHQLQ   309
RHW69169.1       232  GVDHTIDVLVQKGYERSMDSIAAAGLNPIPVQWSEMDKLGISMRAAVLPLKFFK---TNTSGMLSRNKSRGTRWQTHQLK   308
KAG8344931.1     233  GVDHTTDVLVQKGYERSMDSIAAAGLNPIPVQWSEMDKLGISMRAAVLPLKFFK---TNTSGMLSRNKSRGTRWQTHQLK   309
KAH9578299.1     233  GIDFTTDVLVQKGYERSMDSIAAAGLNPIPVQWSEMDKLGISMRAAVLPLKFLK---ANVGGMLSRNKSRGARWQTHQIP   309
XP_028884646.1   233  GVDFTTDVLVQKGYDRSMDSIAAAGLNPIPVQWSEMDKLGISMRAAVLPLKFLK---ANVGGMLSRNKSRGARWQTHQIT   309
XP_029237384.1   233  GVDFTTDVLVQKGYERSMDSVAAAGLNPIPVQWSEMDKLGISMRAAVLPLKFLK---ANVGGMLSRNKSRGSRWQTTQIP   309
XP_811845.1      233  GVDFTTDVLVQKGYERSMDSIAAAGLNPIPVQWSEMDKLGVSMRAAVLPLKFLK---ANVGGMLSRNKSRGTRWQTHQIP   309
KAG5495086.1     251  GVDYTTDVLVQKGFERSMDTLAAAGLNPIAVQWSEMDKLGISMRSAVLPLKFLQspsSTGGGLLQRSRNRGNRWTASQIA   330
XP_010702224.1   252  GIDYTPDVLVQKGFERSMDTLAAAGLNPISVQWSEMDKLGISMRSTVLPLKFLQspsST-GLLQRSRNRVNRWAASQIT   330
XP_015655064.1   248  GVDDLSVDVLVQKGFERSMDALAAAGLNPISVQWSEMDKLGVSMRSAVLPLKFFQspsSS-GGVLQRSRSRVGRWQSNQLA   326
```

```
XP_823393.1      310  K   310
RHW69169.1       309  K   309
KAG8344931.1           -
KAH9578299.1           -
XP_028884646.1         -
XP_029237384.1         -
XP_811845.1            -
KAG5495086.1     331  Q   331
XP_010702224.1   331  Q   331
XP_015655064.1   327  K   327
```

**B**

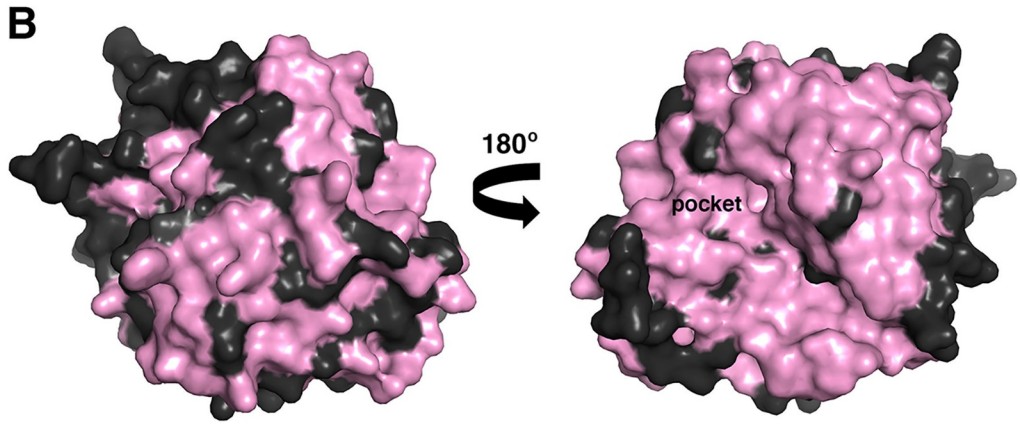

180°

pocket

**Fig 5. Multiple sequence alignment of putative RESC5 homologs and conserved surface residues.** Homolog IDs are indicated to the left. Secondary structural information from the crystal structure is shown above sequence blocks and key residues that interact with RESC6 are boxed and the interface they mediate contacts with (1 or 2) are indicated under the alignment. The proteins are: XP_823393.1; hypothetical protein, conserved [*Trypanosoma brucei brucei* TREU927; RESC5], the protein under study, RHW69169.1; Mitochondrial RNA binding protein [*Trypanosoma brucei equiperdum*]. KAG8344931.1; putative amidinotransferase [*Trypanosoma vivax*], KAH9578299.1; hypothetical protein LSM04_001006 [*Trypanosoma melophagium*], XP_028884646.1 putative amidinotransferase [*Trypanosoma theileri*], XP_029237384.1; putative amidinotransferase [*Trypanosoma rangeli*], XP_811845.1; hypothetical protein [*Trypanosoma cruzi strain* CL Brener], KAG5495086.1; hypothetical protein JKF63_02139 [*Porcisia hertigi*], XP_010702224.1; mitochondrial RNA binding protein, putative [*Leishmania panamensis*], XP_015655064.1 putative mitochondrial RNA binding protein [*Leptomonas pyrrhocoris*]. **B** surface representation of the RESC5 structure colored according to conservation (from the alignment in Fig 5A). Pink regions represent highly conserved regions while dark grey are not conserved. Shown are two "sides" of the molecule. The pocket corresponding to the active site pocket in DDAH is labeled "pocket" and is conserved.

emerged as the central platform that enables processive and efficient editing and utilization of gRNAs. The RESC comprises an RNA-independent core complex of six central proteins, including the RESC1-RESC2 subcomplex that binds and stabilizes all gRNAs [25, 27, 28]. The RESC component RESC5 is required for stable association of RESC1-RESC2 with the core and proper RESC assembly. Thus, RESC5 appears to form a key part of the foundation of the core RESC. However, RESC5, like the other RESC proteins, harbors no identifiable protein motif or fold and to date, no structures have been reported for any of these components. To begin to gain insight into RESC proteins, we performed structural and biochemical studies on the *T. brucei* RESC5 component.

Here we report the structure of the key RESC component, RESC5, to high resolution. The structure revealed that RESC5 has a fold that is utilized in interacting with proteins or protein residues. Specifically, our 1.95 Å structure of the *T. brucei* RESC5 protein revealed that it showed significant structural similarity to the DDAH family of enzymes. DDAH enzymes

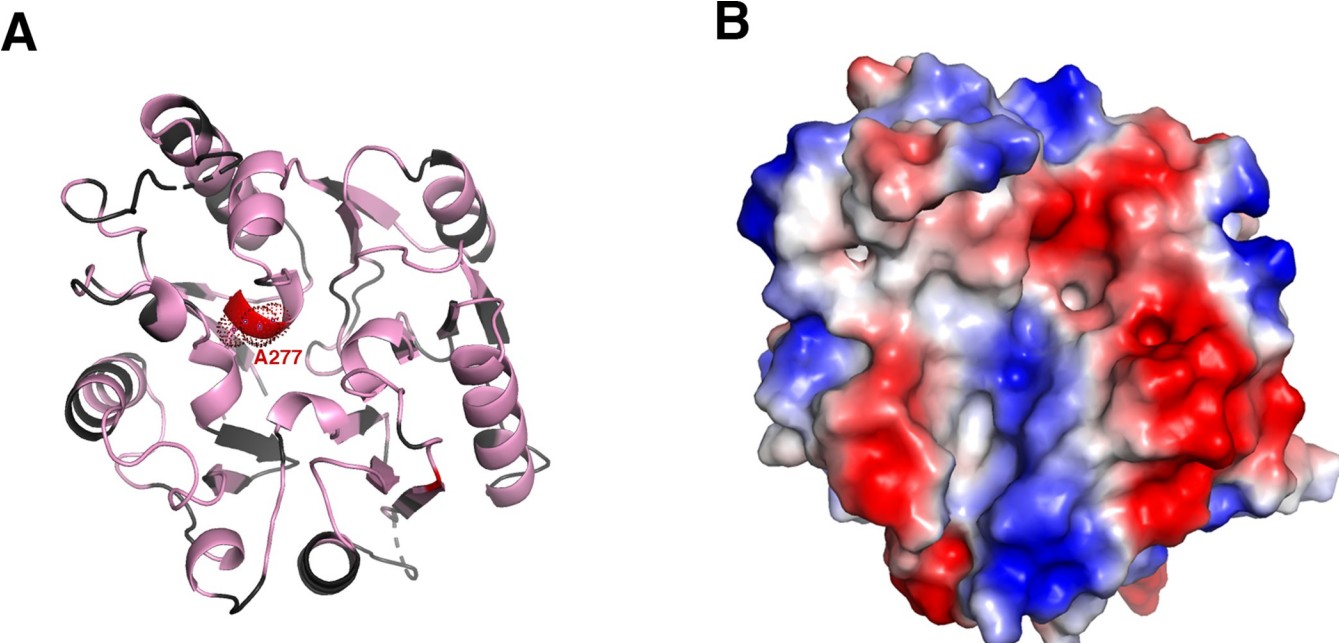

**Fig 6. Electrostatic surface potential of RESC5 reveals potential nucleic acid binding patches. A** RESC5 displayed as a ribbon diagram with the residues corresponding to the catalytic cysteine in DDAH enzymes (an alanine in RESC5) colored red. **B** RESC5 shown as an electrostatic surface representation with blue and red regions representing positive and negative regions, respectively.

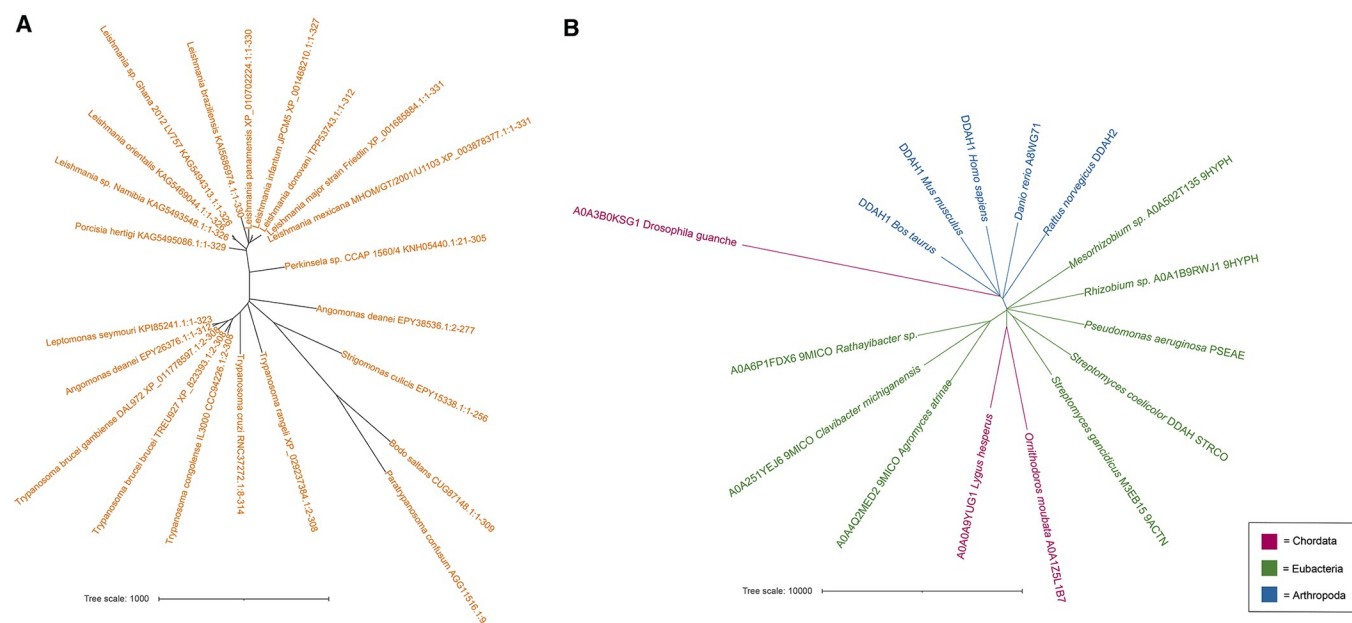

**Fig 7. Phylogenetic trees for RESC5 and DDAH. A** Phylogenetic tree for the RESC5 proteins. The species names are indicated in each branch and the timescale (1000 years) for the branches are included. **B** Phylogenetic tree for DDAH. DDAH proteins were only identified in Chordata (magenta), Eubacteria (green) and Arthropoda (blue). Representative species in each phyla are indicated.

catalyze the hydrolysis of dimethylarginine residues to L-citrulline and L-dimethylamine [38–43]. Dimethylarginine residues are generated by protein degradation and inhibit nitric oxide production by binding to nitric oxide synthase. Hence, DDAH is important in preventing methylarginines from accumulating and inhibiting the generation of nitric oxide [40–42]. While RESC5 harbors the same fold overall as DDAH enzymes, it is missing two essential catalytic residues, the cysteine that functions as the nucleophile to initiate hydrolysis and the histidine, which plays a central role in protonation of the leaving group. In RESC5 these residues are instead an alanine and proline, respectively. But interestingly, RESC5 does harbor most of the active site residues that are involved in substrate binding.

While we found no evidence for RESC5 binding to small molecule methylarginine or citrulline, given the striking conservation of substrate binding residues it is possible that RESC5 may be involved in binding to methylated arginine containing proteins. This represents a particularly intriguing possibility because methylarginine has been identified in several kRNA editing factors that coordinate with the RESC. These include RESC13 (also calledTbRGG2), RESC8 (MRB10130) and RESC12 (MRB4160) [56]. Such interactions could function to modulate the dynamic interactions of the RESC core and associated RESC proteins. Indeed, studies have suggested that interactions in the RESC complex are dynamic and may change during RNA processing events [57]. On the other hand, we noted that the putative pocket in RESC5 harbors an overall positive charge. Hence, the region additionally or alternatively could play a role in RNA or nucleic acid binding. Future studies on RESC5 in the context of the full length RESC complex will be needed to fully elucidate the protein-protein and protein-nucleic acid interactions mediated by RESC5.

In summary, we report the high-resolution crystal structure of a key component of the RESC platform that is essential in kinetoplastid RNA editing.

## Supporting information

**S1 Fig. Uncropped gel shown in Fig 1A.** Shown are the fractions obtained from the SEC analyses of RESC5 (fractions 27–29) in the last three protein lanes, see Fig 1A.
(PDF)

**S2 Fig. Purification and composite omit electron density map for RESC5.** A Purification of RESC5 showing SDS PAGE analyses of fractions collected from Cobalt NTA purification. The top labels indicate imidazole concentrations used to elute the given fractions. **B** Sections of simulated annealing composite omit map calculated in Phenix for the RESC5 structure and contoured at 1σ.
(PDF)

**S3 Fig. Overlay of RESC5 onto the *M. jannaschii* IF6.** RESC5 (pink) was superimposed into the IF6 (pdb code: 1G61) structure (cyan) resulting in an rmsd of 3.5 Å for 1170 Cα atoms.
(PDF)

**S4 Fig. Assessment of putative pockets in the RESC5 structure by CavityPlus.** Analysis of putative pockets (right). The only significant pocket (checked in this list) with a druggability score of 312 is that corresponding to the catalytic site in DDAH enzymes. The pocket is shown superimposed on the RESC5 cartoon at the left.
(PDF)

**S5 Fig. The RESC5(R71A-P180H-A277C) mutant does not bind DDAH substrate or product.** Ribbon diagram showing the overlay of RESC5 (colored pink) with DDAH (colored cyan) and residues subjected to mutation for biochemical assays. Also shown as stick is the location of where the product citrulline binds in the DDAH enzyme.
(PDF)

**S1 File.**
(PDF)

## Acknowledgments

We acknowledge beamline 5.0.2 and 5.0.1 for X-ray diffraction data collection.

## Author Contributions

**Conceptualization:** Raul Salinas, Emily Cannistraci, Maria A. Schumacher.

**Data curation:** Maria A. Schumacher.

**Formal analysis:** Raul Salinas, Emily Cannistraci, Maria A. Schumacher.

**Funding acquisition:** Maria A. Schumacher.

**Investigation:** Raul Salinas, Emily Cannistraci, Maria A. Schumacher.

**Supervision:** Maria A. Schumacher.

**Validation:** Raul Salinas, Emily Cannistraci, Maria A. Schumacher.

**Writing – original draft:** Maria A. Schumacher.

**Writing – review & editing:** Raul Salinas, Emily Cannistraci, Maria A. Schumacher.

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
