## [Decision Letter · Decision Letter 0]

28 Nov 2022

PONE-D-22-27151Structure of the T. brucei kinetoplastid RNA editing substrate-binding complex core component, RESC5PLOS ONE

Dear Dr. Schumacher,

Thank you for submitting your manuscript to PLOS ONE. After careful consideration, we feel that it has merit but does not fully meet PLOS ONE’s publication criteria as it currently stands. Therefore, we invite you to submit a revised version of the manuscript that addresses the points raised during the review process. As you can see from the reviewers comments, there are quite a number of changes that need to be made to the text.

We look forward to receiving your revised manuscript.

Kind regards,

Alexander F. Palazzo, Ph.D.

Academic Editor

PLOS ONE

Journal Requirements:

"This research was supported by Nanaline H Duke Endowed Chair and National Institutes of Health grants (R35GM130290 to M.A.S.). https://www.nigms.nih.gov."

7. We note that you have included the phrase “data not shown” in your manuscript. Unfortunately, this does not meet our data sharing requirements. PLOS does not permit references to inaccessible data. We require that authors provide all relevant data within the paper, Supporting Information files, or in an acceptable, public repository. Please add a citation to support this phrase or upload the data that corresponds with these findings to a stable repository (such as Figshare or Dryad) and provide and URLs, DOIs, or accession numbers that may be used to access these data. Or, if the data are not a core part of the research being presented in your study, we ask that you remove the phrase that refers to these data.

Reviewers' comments:

Reviewer's Responses to Questions

**Comments to the Author**

1. Is the manuscript technically sound, and do the data support the conclusions?

Reviewer #1: Partly

Reviewer #2: Yes

2. Has the statistical analysis been performed appropriately and rigorously? 

Reviewer #1: N/A

Reviewer #2: Yes

3. Have the authors made all data underlying the findings in their manuscript fully available?

Reviewer #1: Yes

Reviewer #2: Yes

4. Is the manuscript presented in an intelligible fashion and written in standard English?

Reviewer #1: Yes

Reviewer #2: Yes

5. Review Comments to the Author

Reviewer #1: Overall, this paper presents a high resolution structure of a RESC core component (RESC5) at 1.95A; which shows a canonical beta-propeller morphology but otherwise it represents a novel structure that shares no identifiable fold/motif to other protein or structures. While the structural data is sound and improvement can be made in its visual presentation (see suggestions below), the functional data is comparatively weak and overreaching in some concluding statements. There are some discrepancies between the in text and figure references that should also be corrected. This manuscript would benefit from more extensive biochemical experiments to probe RESC5 function or substrate binding (for example - could the alanine / proline in RESC5 be mutated to the cysteine / histidine combination to show binding to L-citrulline since the authors suggest that the other catalytic residues are conserved otherwise). While the high resolution RESC5 structure is novel and significant to understanding the overall RESC super complex, I cannot recommend publication without some major revisions to the functional analysis or more validating biochemical experiments.

pg3 - line 12 - 14; please rewrite for clarity, insert reference for pan-editing

pg5 - line 17 & 22; what program was used to predict disorder? please indicate

pg5 - line 17, semantics on "artificial gene"; should either be a gene synthesis product or artificial gene synthesis, please update

pg5 - line 24, RESC5(7-286) - clarify what do you mean by FL expression construct? Full length? It's not full length if it's already N/C-terminally truncated? Please clarify

pg9 - line 19-22; suggest depicting this as an additional inset for Figure 2, aligning sequence with the secondary structure information (like Fig 5), instead of listing it in body of text.

pg10 - line 1; "two to three-stranded"  "two- or three-stranded"

pg10 - line 12, 16; how were the structures superimposed? sequence alignment? structural alignment? please clarify what method/algorith was used (if in Pymol)

pg10 - line 14-19 - how weak is this homology (relative to RESC and DDAH) and why is it relevant than if the homology is significantly weaker as evident in the RMSD/Z-score

pg10 - line 16 (Figure S2); is the superimposition with 165 C-alpha (in-text) or 1170 C-alpha (in figure)? Please clarify.

pg11 - line 8-10 - Fig3A doesn't show this, do you mean Figure 3 itself or Figure 3B?

pg11 - line 24 - Figure 3B not Figure 3A is where L-citrulline is modeled

pg12/pg13 - "phylogenetic analyses of RESC5 and DDAH enzymes" - this section needs some major revision in what it is trying to convey. There's no experimental evidence in this work to suggest that RESC5 and DDAH are evolutionary or functional homolog, they are structural homologs at best (and even then, I don't know if I would say its a striking homology). The concluding statements in this section is hand-wavy at best.

Figure 3 (Fig S2) - show 90' rotation as well (overhead view) to better demonstrate superimposition is actually meaningful

Figure 4, X-axis label is partially cut off on the bottom end

Figure 5B - if you're using blue/yellow for conservation and differences in the sequence alignment (Figure 5A), why not use the same color scheme for the structures in 5b?

Reviewer #2: Schumacher et al report the crystal structure of the T. brucei RESC5 protein (aka MRB11870) which is a member of a multiprotein complex that functions in RNA editing. They determined the structure at a resolution of 1.95 Å of a 279 amino acid soluble recombinant protein that represents much (residues 7-286) of this 310 amino acid protein. The protein eluted as a single peak consistent with the size of a monomer but the crystals had two superimposable monomers that differed by two surface exposed loops. The key finding is that the monomer that was discussed has one domain with a propeller-like fold of 13 beta-strands and 10 helices that has homology to a dimethylarginine dimethylaminohydrolase-like (DDAH) fold. A cavity search of the fold identified pockets corresponding to the active site of DDAH enzymes but the fold lacked residues that are essential for DDAH catalytic function. Functional assays of DDAH substrate or L-citrulline product binding that report a thermal shift in a control experiment with RESC5 and no interaction in the presence of citrulline are inconclusive and are somewhat confusing as written. The authors imply that RESC5 may bind to methylated arginine containing proteins and that basic patches on the protein may function in nucleic acid binding and thus RESC5 could be an editing "factor". They also report weak homology to ribosome anti-association factor IF6 and that RESC5 homologs are only conserved among kinetoplastids although DDAD enzyme homologs are conserved among higher eukaryotes.

Determination of the RESC5 structure is well done and useful and the identification of the DDAH motif is intriguing but the studies of its potential function are uninformative especially given the genetic tractability of T. brucei. Thus, the report does not substantially advance the understanding of its role in RNA editing or its interaction with RESC. There is also no support that it might be a drug target other than its presence in the parasite and absence in the host which however contains DDAD homologs which may present toxicity/off target complications.

6. PLOS authors have the option to publish the peer review history of their article (what does this mean?). If published, this will include your full peer review and any attached files.

Reviewer #1: No

Reviewer #2: No

---

## [Author Response · Author response to Decision Letter 0]

27 Dec 2022

PONE-D-22-27151

Response to Reviewers

Our response to the editor’s and reviewer’s critiques are below (in red text). 

https://journals.plos.org/plosone/s/file?id=wjVg/PLOSOne_formatting_sample_main_body.pdf, 

RESPONSE: We have tried to ensure that we have used the PLOS ONE style requirements in the revision. 

RESPONSE: We have followed this template outline for the document 

"This research was supported by Nanaline H Duke Endowed Chair and National Institutes of Health grants (R35GM130290 to M.A.S.). https://www.nigms.nih.gov."

Please provide an amended statement that declares *all* the funding or sources of support (whether external or internal to your organization) received during this study, as detailed online in our guide for authors at http://journals.plos.org/plosone/s/submit-now. Please also include the statement “There was no additional external funding received for this study.” in your updated Funding Statement. Please include your amended Funding Statement within your cover letter. We will change the online submission form on your behalf.

RESPONSE: We have modified the Funding Statement as recommended and also included this amended Funding Statement in our cover letter. 

RESPONSE: We have fixed this discrepancy. 

RESPONSE: We have done so.

RESPONSE: Our data availability statement included the information about the deposition. Specifically, the coordinates and structure factor amplitudes for the RESC5 structure have been deposited in the Protein Data Bank under the accession code 8DPK and are currently HPUB (hold for publication). They will be released and made available upon manuscript acceptance. 

RESPONSE: We have added the uncropped gel figure shown in Fig 1 in the supplementary file. The other gel (Fig S2 is shown in uncropped form). 

RESPONSE: We have noted this in the Cover letter.

7. We note that you have included the phrase “data not shown” in your manuscript. Unfortunately, this does not meet our data sharing requirements. PLOS does not permit references to inaccessible data. We require that authors provide all relevant data within the paper, Supporting Information files, or in an acceptable, public repository. Please add a citation to support this phrase or upload the data that corresponds with these findings to a stable repository (such as Figshare or Dryad) and provide and URLs, DOIs, or accession numbers that may be used to access these data. Or, if the data are not a core part of the research being presented in your study, we ask that you remove the phrase that refers to these data.

RESPONSE: The data were not a core part of the research and hence we have removed the phrase referring to these data. 

Reviewers' comments:

Reviewer's Responses to Questions

Comments to the Author

1. Is the manuscript technically sound, and do the data support the conclusions?

Reviewer #1: Partly

Reviewer #2: Yes

2. Has the statistical analysis been performed appropriately and rigorously? 

Reviewer #1: N/A

Reviewer #2: Yes

3. Have the authors made all data underlying the findings in their manuscript fully available?

Reviewer #1: Yes

Reviewer #2: Yes

4. Is the manuscript presented in an intelligible fashion and written in standard English?

Reviewer #1: Yes

Reviewer #2: Yes

5. Review Comments to the Author

Reviewer #1: Overall, this paper presents a high resolution structure of a RESC core component (RESC5) at 1.95A; which shows a canonical beta-propeller morphology but otherwise it represents a novel structure that shares no identifiable fold/motif to other protein or structures. While the structural data is sound and improvement can be made in its visual presentation (see suggestions below), the functional data is comparatively weak and overreaching in some concluding statements. There are some discrepancies between the in text and figure references that should also be corrected. This manuscript would benefit from more extensive biochemical experiments to probe RESC5 function or substrate binding (for example - could the alanine / proline in RESC5 be mutated to the cysteine / histidine combination to show binding to L-citrulline since the authors suggest that the other catalytic residues are conserved otherwise). While the high resolution RESC5 structure is novel and significant to understanding the overall RESC super complex, I cannot recommend publication without some major revisions to the functional analysis or more validating biochemical experiments.

RESPONSE: As suggested, we generated a RESC5 mutant in which key catalytic residues within the RESC5 pocket were changed to corresponding residues in DDAH. The mutant protein was expressed, purified and utilized in thermal shift assays. The data are now included as Fig 4B (also see Fig S5). In addition, we have followed the reviewer’s suggestions in presentation and text edits, as detailed below. We thank the reviewer for their very helpful critiques and suggestions. 

pg3 - line 12 - 14; please rewrite for clarity, insert reference for pan-editing

RESPONSE: We have described pan editing for clarity and have inserted references at this point. We thank the reviewer for suggesting this.

pg5 - line 17 & 22; what program was used to predict disorder? please indicate

RESPONSE: The program used was GOR version IV. This has been added along with the relevant reference. 

pg5 - line 17, semantics on "artificial gene"; should either be a gene synthesis product or artificial gene synthesis, please update

RESPONSE: We have changed this to “synthetic gene”. 

pg5 - line 24, RESC5(7-286) - clarify what do you mean by FL expression construct? Full length? It's not full length if it's already N/C-terminally truncated? Please clarify

RESPONSE: This has been corrected to the construct encoding RESC5(7-310). This construct contains all the C-terminal residues but lacks the N-terminal residues 1-6.

pg9 - line 19-22; suggest depicting this as an additional inset for Figure 2, aligning sequence with the secondary structure information (like Fig 5), instead of listing it in body of text.

RESPONSE:We would prefer to retain these details in the text. 

pg10 - line 1; "two to three-stranded"  "two- or three-stranded"

RESPONSE:Changed to “or” as suggested. 

pg10 - line 12, 16; how were the structures superimposed? sequence alignment? structural alignment? please clarify what method/algorith was used (if in Pymol)

RESPONSE:The structures were superimposed using the program Secondary-structure matching (SSM), which is a tool used for protein structure alignment in three dimensions. This information and the reference [50] have been included in the revision. 

pg10 - line 14-19 - how weak is this homology (relative to RESC and DDAH) and why is it relevant than if the homology is significantly weaker as evident in the RMSD/Z-score.

RESPONSE: The RMSD between RESC5 and IF6 is 5.8 A for 165 Ca atoms whereas the RMSD with DDAH structure is 2.4 A for a superimposition of 247 Ca atoms. So the DDAH structures are much more similar to RESC5 than IF6. However, we included IF6 as it was output in our structure similarity searches. 

pg10 - line 16 (Figure S2); is the superimposition with 165 C-alpha (in-text) or 1170 C-alpha (in figure)? Please clarify.

RESPONSE: We thank the reviewer for catching this. This was a typo and should have been 165 Ca atoms. This has been fixed in the Fig. S2 legend. 

pg11 - line 8-10 - Fig3A doesn't show this, do you mean Figure 3 itself or Figure 3B?

RESPONSE: Yes, this should be Fig 3B. This has been corrected now in the text. We thank the reviewer for pointing this out.

pg11 - line 24 - Figure 3B not Figure 3A is where L-citrulline is modeled

RESPONSE:Again, this should have been Fig 3B. this has been fixed and we thank the reviewer for catching this.

pg12/pg13 - "phylogenetic analyses of RESC5 and DDAH enzymes" - this section needs some major revision in what it is trying to convey. There's no experimental evidence in this work to suggest that RESC5 and DDAH are evolutionary or functional homolog, they are structural homologs at best (and even then, I don't know if I would say its a striking homology). The concluding statements in this section is hand-wavy at best.

RESPONSE: As suggested, we have rewritten and heavily revised this section of the manuscript. The reviewer is correct, we have no experimental evidence to suggest that RESC5 and DDAH are evolutionarily related. We feel it is important to describe/determine whether DDAH and RESC5 protein homologs are found in all eukaryotes or are limited to certain organisms. Indeed, an interesting finding is that RESC5 protein homologs are only found in the kinetoplastida while these organisms have no DDAH homologs. We have rewritten the section to simply report this information and reflect these findings. 

Figure 3 (Fig S2) - show 90' rotation as well (overhead view) to better demonstrate superimposition is actually meaningful

RESPONSE: We have added the 90 degree rotated view (overhead view), as recommended. 

Figure 4, X-axis label is partially cut off on the bottom end

RESPONSE: Thank the reviewer for pointing this out. We have made sure that the X-axis is not cut off in the uploaded figure. 

Figure 5B - if you're using blue/yellow for conservation and differences in the sequence alignment (Figure 5A), why not use the same color scheme for the structures in 5b?

RESPONSE: We have matched the colors by making the conservation pink and non-conserved grey in both the alignment and the models. We thank the reviewer for this suggestion.

Reviewer #2: Schumacher et al report the crystal structure of the T. brucei RESC5 protein (aka MRB11870) which is a member of a multiprotein complex that functions in RNA editing. They determined the structure at a resolution of 1.95 Å of a 279 amino acid soluble recombinant protein that represents much (residues 7-286) of this 310 amino acid protein. The protein eluted as a single peak consistent with the size of a monomer but the crystals had two superimposable monomers that differed by two surface exposed loops. The key finding is that the monomer that was discussed has one domain with a propeller-like fold of 13 beta-strands and 10 helices that has homology to a dimethylarginine dimethylaminohydrolase-like (DDAH) fold. A cavity search of the fold identified pockets corresponding to the active site of DDAH enzymes but the fold lacked residues that are essential for DDAH catalytic function. Functional assays of DDAH substrate or L-citrulline product binding that report a thermal shift in a control experiment with RESC5 and no interaction in the presence of citrulline are inconclusive and are somewhat confusing as written. The authors imply that RESC5 may bind to methylated arginine containing proteins and that basic patches on the protein may function in nucleic acid binding and thus RESC5 could be an editing "factor". They also report weak homology to ribosome anti-association factor IF6 and that RESC5 homologs are only conserved among kinetoplastids although DDAD enzyme homologs are conserved among higher eukaryotes.

Determination of the RESC5 structure is well done and useful and the identification of the DDAH motif is intriguing but the studies of its potential function are uninformative especially given the genetic tractability of T. brucei. Thus, the report does not substantially advance the understanding of its role in RNA editing or its interaction with RESC. There is also no support that it might be a drug target other than its presence in the parasite and absence in the host which however contains DDAD homologs which may present toxicity/off target complications.

RESPONSE: We thank the reviewer for their comments and suggestions. As recommended, we have rewritten the section describing the thermal shift assays. We mentioned RESC5 as a drug target in the discussion, but the reviewer’s point is well taken- we have now removed that sentence. Our focus is on the determination and description of the first structure of any RESC component to high resolution. However, as recommended by the reviewer we have performed additional experiments to assess the structure and function of RESC5. These data are reported in Fig 4B. 

6. PLOS authors have the option to publish the peer review history of their article (what does this mean?). If published, this will include your full peer review and any attached files.

Do you want your identity to be public for this peer review? For information about this choice, including consent withdrawal, please see our Privacy Policy.

Reviewer #1: No

Reviewer #2: No

While revising your submission, please upload your figure files to the Preflight Analysis and Conversion Engine (PACE) digital diagnostic tool, https://pacev2.apexcovantage.com/. PACE helps ensure that figures meet PLOS requirements. To use PACE, you must first register as a user. Registration is free. Then, login and navigate to the UPLOAD tab, where you will find detailed instructions on how to use the tool. If you encounter any issues or have any questions when using PACE, please email PLOS at figures@plos.org. Please note that Supporting Information files do not need this step. We have done this with the result being that all files were found to be “converted to a valid TIF file”, RGB, 8 bit depth, with LZW compression. Files that were too large were resized, while retaining the needed resolution and resulting in sizes of 10 MB or less.

---

## [Editor Report · Decision Letter 1]

3 Jan 2023

PONE-D-22-27151R1Structure of the T. brucei kinetoplastid RNA editing substrate-binding complex core component, RESC5PLOS ONE

Dear Dr. Schumacher,

Thank you for submitting your manuscript to PLOS ONE. After careful consideration, we feel that it has merit but that the manuscript requires a text change before it fully meets PLOS ONE’s publication criteria. Therefore, we invite you to submit a revised version of the manuscript that addresses the points raised during the review process. In particular, we ask that you please omit the term "PCR" from the manuscript when you are referring to protocols where a thermocycler is used to measure protein stability. "PCR" stands for the polymerase chain reaction and involves the amplification of DNA using oligonucleotides and a thermo-stable polymerase. For an example on how to correctly write about the method you use, see: https://pubmed.ncbi.nlm.nih.gov/23046506/

We look forward to receiving your revised manuscript.

Kind regards,

Alexander F. Palazzo, Ph.D.

Academic Editor

PLOS ONE
---

## [Author Response · Author response to Decision Letter 1]

4 Jan 2023

We have removed all funding related text from the manuscript.

 (we also include in the response to reviewers, our responses to the comments from the reviewers).

---

## [Editor Report · Decision Letter 2]

8 Feb 2023

Structure of the T. brucei kinetoplastid RNA editing substrate-binding complex core component, RESC5

PONE-D-22-27151R2

Dear Dr. Schumacher,

We’re pleased to inform you that your manuscript has been judged scientifically suitable for publication and will be formally accepted for publication once it meets all outstanding technical requirements.

Kind regards,

Alexander F. Palazzo, Ph.D.

Academic Editor

PLOS ONE
---

## [Editor Report · Acceptance letter]

17 Feb 2023

PONE-D-22-27151R2 

Structure of the *T. brucei kinetoplastid* RNA editing substrate-binding complex core component, RESC5 

Dear Dr. Schumacher:

I'm pleased to inform you that your manuscript has been deemed suitable for publication in PLOS ONE. Congratulations! Your manuscript is now with our production department. 

Kind regards, 

on behalf of

Dr. Alexander F. Palazzo 

Academic Editor

PLOS ONE